# Artificial sweeteners and cancer risk: Results from the NutriNet-Santé population-based cohort study

Charlotte Debras[1,2]*, Eloi Chazelas[1,2], Bernard Srour[1,2], Nathalie Druesne-Pecollo[1,2], Younes Esseddik[1], Fabien Szabo de Edelenyi[1], Cédric Agaësse[1], Alexandre De Sa[1], Rebecca Lutchia[1], Stéphane Gigandet[3], Inge Huybrechts[2,4], Chantal Julia[1,5], Emmanuelle Kesse-Guyot[1,2], Benjamin Allès[1], Valentina A. Andreeva[1], Pilar Galan[1,2], Serge Hercberg[1,2,5], Mélanie Deschasaux-Tanguy[1,2], Mathilde Touvier[1,2]

1 Sorbonne Paris Nord University, INSERM U1153, INRAE U1125, CNAM, Nutritional Epidemiology Research Team (EREN), Epidemiology and Statistics Research Center, University of Paris (CRESS), Bobigny, France, 2 French Network for Nutrition and Cancer Research (NACRe network), Jouy-en-Josas, France, 3 Open Food Facts, Saint-Maur-des-Fossés, France, 4 International Agency for Research on Cancer, World Health Organization, Lyon, France, 5 Public Health Department, Avicenne Hospital, Assistance Publique–Hôpitaux de Paris, Bobigny, France

* c.debras@eren.smbh.univ-paris13.fr

**Data Availability Statement:** Researchers from public institutions can submit a collaboration request including information on the institution and a brief description of the project to

## Abstract

### Background

The food industry uses artificial sweeteners in a wide range of foods and beverages as alternatives to added sugars, for which deleterious effects on several chronic diseases are now well established. The safety of these food additives is debated, with conflicting findings regarding their role in the aetiology of various diseases. In particular, their carcinogenicity has been suggested by several experimental studies, but robust epidemiological evidence is lacking. Thus, our objective was to investigate the associations between artificial sweetener intakes (total from all dietary sources, and most frequently consumed ones: aspartame [E951], acesulfame-K [E950], and sucralose [E955]) and cancer risk (overall and by site).

### Methods and findings

Overall, 102,865 adults from the French population-based cohort NutriNet-Santé (2009–2021) were included (median follow-up time = 7.8 years). Dietary intakes and consumption of sweeteners were obtained by repeated 24-hour dietary records including brand names of industrial products. Associations between sweeteners and cancer incidence were assessed by Cox proportional hazards models, adjusted for age, sex, education, physical activity, smoking, body mass index, height, weight gain during follow-up, diabetes, family history of cancer, number of 24-hour dietary records, and baseline intakes of energy, alcohol, sodium, saturated fatty acids, fibre, sugar, fruit and vegetables, whole-grain foods, and dairy products. Compared to non-consumers, higher consumers of total artificial sweeteners (i.e., above the median exposure in consumers) had higher risk of overall cancer ($n$ = 3,358 cases, hazard ratio [HR] = 1.13 [95% CI 1.03 to 1.25], $P$-trend = 0.002). In particular,

collaboration@etude-nutrinet-sante.fr. All requests will be reviewed by the steering committee of the NutriNet-Santé study. If the collaboration is accepted, a data access agreement will be necessary and appropriate authorizations from the competent administrative authorities may be needed. In accordance with existing regulations, no personal data will be accessible.

**Funding:** The NutriNet-Santé study was supported by the following public institutions: Ministère de la Santé, Santé Publique France, Institut National de la Santé et de la Recherche Médicale (INSERM), Institut national de recherche pour l'agriculture, l'alimentation et l'environnement (INRAE), Conservatoire National des Arts et Métiers (CNAM) and Université Sorbonne Paris Nord. CD was supported by a grant from the French National Cancer Institute (INCa, grant #2019-158). EC was supported by a Doctoral Fellowship from Université Sorbonne Paris Nord to Galilée Doctoral School. This project has received funding from the European Research Council (ERC) under the European Union's Horizon 2020 research and innovation program (grant agreement No 864219), the French National Cancer Institute (INCa_14059), the French Ministry of Health (arrêté 29.11.19) and the IdEx Université de Paris (ANR-18-IDEX-0001). This project was awarded the NACRe (French network for Nutrition And Cancer Research) Partnership Label. Researchers were independent from funders. Funders had no role in the study design, the collection, analysis, and interpretation of data, the writing of the report, and the decision to submit the article for publication.

**Competing interests:** I have read the journal's policy and the authors of this manuscript have the following competing interests: SG is co-founder of Open Food Facts, a non-profit project developed by thousands of volunteers from around the world. It is a free and open-data food product database designed to help citizens make better food choices.

**Abbreviations:** ADI, acceptable daily intake; ASB, artificially sweetened beverage; BMI, body mass index; CI, confidence interval; EFSA, European Food Safety Authority; HR, hazard ratio; IPAQ, International Physical Activity Questionnaire; WHO, World Health Organization.

aspartame (HR = 1.15 [95% CI 1.03 to 1.28], $P$ = 0.002) and acesulfame-K (HR = 1.13 [95% CI 1.01 to 1.26], $P$ = 0.007) were associated with increased cancer risk. Higher risks were also observed for breast cancer ($n$ = 979 cases, HR = 1.22 [95% CI 1.01 to 1.48], $P$ = 0.036, for aspartame) and obesity-related cancers ($n$ = 2,023 cases, HR = 1.13 [95% CI 1.00 to 1.28], $P$ = 0.036, for total artificial sweeteners, and HR = 1.15 [95% CI 1.01 to 1.32], $P$ = 0.026, for aspartame). Limitations of this study include potential selection bias, residual confounding, and reverse causality, though sensitivity analyses were performed to address these concerns.

## Conclusions

In this large cohort study, artificial sweeteners (especially aspartame and acesulfame-K), which are used in many food and beverage brands worldwide, were associated with increased cancer risk. These findings provide important and novel insights for the ongoing re-evaluation of food additive sweeteners by the European Food Safety Authority and other health agencies globally.

## Trial registration

ClinicalTrials.gov NCT03335644.

## Author summary

### Why was this study done?

- The food industry uses artificial sweeteners in a wide range of foods and beverages, as alternatives to added sugars, for which deleterious effects on several chronic diseases are now well established.

- However, the safety of artificial sweeteners is questioned, and their role in the aetiology of various diseases is debated.

- In particular, their carcinogenicity has been suggested by several experimental studies, but robust epidemiological evidence is lacking.

- Previous observational studies have investigated the associations only between cancer risk and the consumption of artificially sweetened beverages, used as a proxy.

### What did the researchers do and find?

- In this large cohort of 102,865 French adults, artificial sweeteners (especially aspartame and acesulfame-K) were associated with increased overall cancer risk (hazard ratio [HR] for higher consumers compared to non-consumers = 1.13 [95% CI 1.03 to 1.25], $P$-trend = 0.002).

- More specifically, aspartame intake was associated with increased breast (HR = 1.22 [95% CI 1.01 to 1.48], $P$ = 0.036) and obesity-related (HR = 1.15 [95% CI 1.01 to 1.32], $P$ = 0.026) cancer risks.

**What do these findings mean?**

- These results suggest that artificial sweeteners, used in many food and beverage brands worldwide, may represent a modifiable risk factor for cancer prevention.

- These findings provide novel information in the context of the ongoing re-evaluation of food additive sweeteners by the European Food Safety Authority and other health agencies globally.

## Introduction

Given the deleterious health effects of excess sugar intake (e.g., weight gain, cardiometabolic disorders, dental caries), the World Health Organization (WHO) recommends limiting sugar consumption to less than 10% of daily energy intake [1]. However, as liking for sweet taste is widespread globally, the food industry started to use artificial sweeteners as alternatives to reduce added sugar content and corresponding calories while maintaining sweetness. In addition, in order to increase palatability, manufacturers include artificial sweeteners in some food products that do not traditionally contain added sugar (e.g., flavoured potato chips). High-intensity sweeteners (hereafter referred to as 'artificial sweeteners') are food additives with high sweetening power yet providing little energy. Aspartame (E951), a well-known artificial sweetener, is found in nearly 1,400 food products on the French market, and more than 6,000 worldwide [2,3]. Its energy value is similar to sugar (4 kcal/g) but its sweetness is 200 times higher [4], meaning a much smaller amount of aspartame is needed for a comparable taste. Other artificial sweeteners are even calorie-free, e.g., acesulfame-K (E950) and sucralose (E955), which are respectively 200 and 600 times sweeter than sucrose [4].

Previous evaluations by health authorities concluded that there was insufficient evidence for risk for the consumption of low- and no-calorie sweeteners under established acceptable daily intakes (ADIs) [4,5]. However, recent epidemiological and experimental studies with conflicting results have reactivated the debate on the safety of these additives. In this context, several health authorities are currently re-evaluating artificial sweeteners, including the European Food Safety Authority (EFSA) [6]. Indeed, while some epidemiological studies did not support the involvement of artificial sweeteners in various health outcomes (e.g., weight loss or weight gain [7–9], glycaemic control [7,8], cardiovascular/kidney diseases [7]), others suggested associations with higher incidence of obesity, hypertension, metabolic syndrome, type 2 diabetes, and cardiovascular events [10].

Regarding cancer, all previous evaluations agreed upon the fact that additional studies, especially in humans, were needed [4]. In particular, experts have urged for a re-evaluation by public health authorities of aspartame's role in cancer development [11,12], based on previous and recent findings in animal models [11,13], in vitro studies [14,15], and, to a lesser extent, human data [2,16]. Findings about other artificial sweeteners also raise questions regarding their potential role in carcinogenesis based on in vivo studies [13,17]. To our knowledge, no previous prospective cohort has investigated the association of cancer risk with quantitative artificial sweetener intakes from all dietary sources, distinguishing the different types of sweeteners. Indeed, so far, human-derived data have mostly investigated artificial sweetener intakes by using the overall consumption of artificially sweetened beverages (ASBs) as a proxy. A more precise assessment of exposure to artificial sweeteners from a broader range of ultra-processed

products (e.g., flavoured yogurts, low-sugar snacks, ready-to-go meals, table-top sweeteners) appears necessary. Besides, since most previous epidemiological studies did not collect data on the brand names of products, data are lacking regarding the specific types of sweeteners consumed by the participants (e.g., aspartame, acesulfame-K, sucralose).

Thus, the objective of our study was to investigate the associations between intakes of artificial sweeteners (total and most consumed ones) and cancer risk (overall and by most frequent cancer sites) in the large-scale population-based NutriNet-Santé cohort, based on detailed dietary data including names/brands of industrial products.

## Methods

### Study population and data collection

The NutriNet-Santé study is a web-based cohort dedicated to investigating the associations between nutrition and health [18]. Enrolment of participants from the French population was initiated in May 2009 and is still ongoing. The NutriNet-Santé volunteers are adults aged ≥18 years with Internet access recruited through extensive multimedia campaigns. Each participant is followed via questionnaires available and regularly added in their personal account on the study website (https://etude-nutrinet-sante.fr). In particular, detailed information is collected at baseline and every year thereafter through a 5-questionnaire kit, regarding health status (e.g., personal and family history of diseases and drug use), anthropometric data (height, weight) [19,20], physical activity (validated 7-day assessment via the International Physical Activity Questionnaire [IPAQ] [21]), lifestyle and sociodemographic characteristics (e.g., sex, date of birth, educational level, occupation, smoking status, number of children) [22], and diet (see below).

An electronic informed consent is provided by each participant. The NutriNet-Santé study, registered at ClinicalTrials.gov (NCT03335644), is conducted according to the Declaration of Helsinki guidelines and is approved by the Institutional Review Board of the French Institute of Health and Medical Research (Inserm) and the Commission Nationale de l'Informatique et des Libertés (CNIL 908450/909216). All methods have been described in line with the Strengthening the Reporting of Observational Studies in Epidemiology–Nutritional Epidemiology guidelines (S1 STROBE-nut Checklist).

The NutriNet-Santé study was developed to investigate the relationships between multiple dietary exposures and the incidence of chronic diseases, such as cancer. The general protocol of the cohort, written in 2008 before the beginning of the study, is available online [23]. Regarding food additives specifically, the present work is part of a series of pre-specified analyses that are included in a project funded by the European Research Council (https://erc. europa.eu/news-events/magazine/erc-2019-consolidator-grants-examples#ADDITIVES).

### Patient involvement statement

The research question developed in this article corresponds to a concern expressed by some participants involved in the NutriNet-Santé cohort, and of the public in general. Participants in the study are thanked in the Acknowledgements. The results of the present study are disseminated to the NutriNet-Santé participants through the cohort website, public seminars, and a press release.

### Dietary assessment

Dietary intakes are collected every 6 months by 3 non-consecutive web-based 24-hour dietary records, randomly assigned over 15 days (2 weekdays and 1 weekend day). Participants declare

all foods and beverages consumed during main meals and other eating occasions, and they provide information on portion sizes via validated photographs or standard serving containers [24]. Baseline dietary intakes were evaluated by averaging all 24-hour dietary records provided during the first 2 years of follow-up (up to 15 records). Daily intakes of energy, alcohol, and macro- and micronutrients were assessed via the NutriNet-Santé food composition table (providing nutritional composition for about 3,500 items) [25]. Nutrient intakes from composite dishes were estimated according to usual French recipes as defined by nutrition professionals. Dietary energy under-reporters were identified using basal metabolic rate and the Goldberg cut-off method [26], and excluded from the analyses. The detailed methodology for identifying under-reporting is presented in Method A in S1 Appendix. The 24-hour dietary records were validated against an interview by a trained dietitian [27] and against blood and urinary biomarkers [28,29].

## Artificial sweetener intakes

Artificial sweetener intakes were assessed through the 24-hour dietary records, in which brands and commercial names of industrial products were routinely collected, enabling us to assess exposure to each food additive. Additive exposure assessment in the NutriNet-Santé cohort has been previously described in detail [30]. Briefly, the presence or absence of each additive in each specific food product consumed was determined using 3 large-scale composition databases: the French food safety agency database Oqali (https://www.oqali.fr/oqali_eng/) [31], Open Food Facts (https://fr-en.openfoodfacts.org/) [3], and Mintel's Global New Products Database [32]. Dynamic matching was applied, meaning that products were matched date-to-date: The date of consumption of each food or beverage declared by each participant was used to match the product to the closest composition data, thus accounting for potential reformulations. In total, quantitative doses of additives were estimated by approximately 2,700 laboratory assays (either specifically performed by our laboratory for this project—89% of the assays regarding artificial sweeteners—or performed by accredited laboratories upon request of the consumer association UFC–Que Choisir) on different food matrices for the main additive–food vector pairs. Quantitative doses were completed with information for generic food categories provided by EFSA and the Joint FAO/WHO Expert Committee on Food Additives (JECFA) [33]. This methodology allowed us to assess exposure for the following artificial sweeteners: acesulfame-K (European food additive identification number E950), aspartame (E951), cyclamates (E952), saccharin (E954), sucralose (E955), thaumatin (E957), neohesperidine dihydrochalcone (E959), steviol glycosides (E960), and salt of aspartame-acesulfame (E962); the quantities consumed of all these artificial sweeteners were summed to calculate the variable 'total artificial sweeteners'. Specific analyses were performed for the most represented artificial sweeteners in the cohort: aspartame, acesulfame-K, and sucralose. All other artificial sweeteners were consumed by less than 3.5% of participants.

## Cancer case ascertainment

Participants are asked to report all medications/treatment and major health events on the annual health questionnaire, a specific check-up questionnaire every 6 months, or at any time on their NutriNet-Santé account. In order to validate reported incident cancer cases, participants were contacted by a physician of the research team to provide any relevant medical and anatomopathological reports. If necessary, the participant's physicians and/or hospitals were also contacted to provide the requested information. All cases reported up to 22 January 2021 were investigated. In addition, the data are linked to the medico-administrative databases of the national health insurance system database (SNIIRAM) and the national mortality registry

(CépiDc) to ensure completeness of morbidity and mortality information and to limit bias associated with unreported cases. Medical information was obtained for more than 90% of incident cases, and 95% of these were validated; therefore, all incident cases declared were included in the present study, unless they were not validated based on the information provided. Cases were then classified using the International Classification of Diseases–10th Revision [34]. In this study, all first primary cancers (except for basal cell carcinoma) diagnosed between inclusion and 22 January 2021 were considered as cases. Obesity-related cancers are all cancers for which obesity is involved in their aetiology as one of the risk (or protective) factors, as recognised by the World Cancer Research Fund (independently of participant BMI status) [35]: colorectal, stomach, liver, mouth, pharynx, larynx, oesophageal, breast (with opposite associations pre- and postmenopause), ovarian, endometrial, and prostate cancers.

## Statistical analysis

Energy under-reporters, as well as those with prevalent cancer at baseline, were excluded. A detailed flowchart is presented in Fig 1.

Since a substantial proportion of the population were non-consumers of artificial sweeteners, participants were divided into 3 groups: non-consumers, lower consumers, and higher consumers, the latter 2 being separated by the sex-specific median of consumption in the study population. Baseline characteristics were examined across categories of total artificial sweetener intake and were compared using ANOVA tests for continuous variables or $\chi^2$ tests for categorical variables. Associations between sweetener intake (all artificial sweeteners, aspartame, acesulfame-K, and sucralose) and cancer risk (overall and by type) were assessed by Cox proportional hazards models with age modelled as the time scale. Specific cancer types considered in this study were breast and prostate (i.e., the most frequent cancer sites in women and men in France [36] and in the cohort) as well as the group of obesity-related cancers. Participants contributed person-time from their inclusion in the cohort until the date of cancer diagnosis, date of last follow-up, date of death, or 22 January 2021, whichever occurred first. Cause-specific hazards were computed so that death and cancer events other than the one

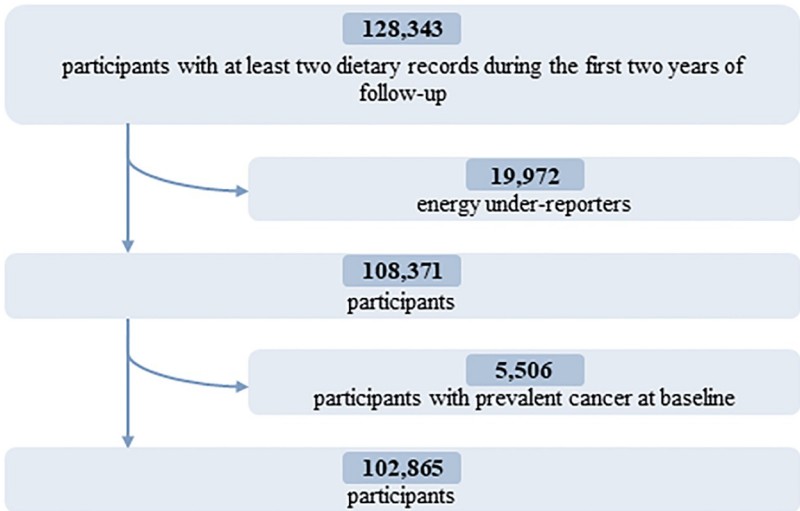

**Fig 1. Flowchart for the selection of the study population: NutriNet-Santé cohort, France, 2009–2021.**

studied (for site-specific analyses) occurring during follow-up were handled as competing risks. The Fine and Gray subdistribution hazard model was also tested in sensitivity analysis. Hazard ratios (HRs) and 95% confidence intervals (CIs) were estimated with the non-consumer group as the reference category. *P*-trend was obtained using the ordinal score for each group (non-consumers: 1; lower consumers: 2; higher consumers: 3). The proportional hazards assumption of the Cox model was confirmed with the rescaled Schoenfeld-type residuals method (Fig A in S1 Appendix). We assessed linearity by comparing the model with the 3 distinct categories of sweetener intake to a model with a linear trend across these categories, using the Akaike information criterion. Consumer (overall) versus non-consumer analyses were also conducted, and this model with 2 categories of exposure was compared to the main model with a formal test for heterogeneity. Missing values for any covariates were handled using the multiple imputation by chained equations (MICE) method [37] (15 imputed datasets) (details in Method B in S1 Appendix). The main analyses were adjusted for the following covariates: sociodemographic characteristics (age [time scale], sex [except for breast and prostate cancer analyses], educational level), lifestyle characteristics (physical activity [IPAQ] [21], smoking status, number of smoked cigarettes in pack-years), anthropometric characteristics (body mass index [BMI], height, percentage weight gain during follow-up), personal and family medical history (prevalent type 1 or type 2 diabetes, family history of cancer), number of 24-hour dietary records, and baseline intakes of energy and food groups/key nutrients for which a direct or indirect role in cancer aetiology has been strongly suggested [35] (alcohol, sodium, saturated fatty acids, fibre, total sugar, fruit and vegetables, whole-grain foods, and dairy products). Breast cancer analyses were additionally adjusted for age at menarche, age at first childbirth, number of biological children, baseline menopausal status, and oral contraceptive use and hormonal treatment for menopause at baseline and during follow-up. Coding for these covariates is indicated in the footnotes to the tables. In analyses specific to 1 artificial sweetener, models were mutually adjusted for the total intake of all other artificial sweeteners. We report minimally adjusted (for age and sex only) and fully adjusted HRs for the associations between artificial sweeteners (total, aspartame, acesulfame-K, and sucralose) and cancer risk (overall, breast, prostate, and obesity-related).

In order to explore the question of which, between sugar and artificial sweeteners, may be more problematic regarding cancer risk, participants were categorised into 6 classes according to their intake levels of artificial sweeteners (non-consumers, lower consumers, and higher consumers) and sugar ($</\geq$ the French recommended limit of 100 g/day [38]). Cancer risks were compared 2-by-2 across the 6 categories, and in particular between the categories 'higher artificial sweetener consumption and sugar intake below the official recommended limit' and 'no artificial sweetener consumption and sugar intake exceeding the recommended limit', with the latter category being the reference category. Further analyses were conducted to investigate the associations of artificial sweetener consumption with premenopausal and postmenopausal breast cancers separately, and menopause-related heterogeneity was assessed using likelihood ratio tests comparing the estimated log-likelihood of a model to that of the same model plus a multiplicative interaction term for menopausal status and the artificial sweetener exposure. Formal interactions between BMI ($</\geq$25 kg/m$^2$) and each artificial sweetener were tested for all studied outcomes by entering the product of the 2 variables into Cox models. Similarly, the 3-way interaction between the 3 main artificial sweeteners (aspartame, acesulfame-K, and sucralose) was tested for all studied outcomes by including the product of the 3 variables into Cox models. The multi-sweetener exposure aspect was explored by classifying artificial sweetener consumers as consumers of 1 type of artificial sweetener, consumers of 2 different sweeteners, and consumers of 3 different sweeteners, and comparing overall cancer risk 2-by-2 across these categories, adjusting for the total dose of artificial sweetener intake.

A series of sensitivity analyses were performed to assess the robustness of the findings, including restriction of the study population to participants with at least four 24-hour dietary records and exclusion of participants with prevalent diabetes. Models with further adjustments for added sugar intake, sugary beverage consumption, proportion of ultra-processed foods in the diet, being on a weight-loss/calorie-restricted diet, and 'healthy' and 'Western' dietary patterns (derived by principal component analysis) instead of food groups, and models with artificial sweetener intakes coded as time-dependent exposure variables, were also tested. Some participants with subclinical cancer may get sick and change their dietary habits during the months preceding their diagnosis. Thus, in order to counter this potential reverse causality bias, we performed a sensitivity analysis with follow-up starting at age at entry into the cohort plus 2 years.

The use of Cox models, the 3-category coding for sweetener exposure, and the adjustment for main confounders (sociodemographic, anthropometric, nutritional, and health-related) were pre-specified. The main analyses added following the review process were as follows: linearity check in relation to the trend across categories, interaction tests between artificial sweetener exposure and menopausal status, heterogeneity tests to compare exposure coding strategies, models restricted to non-smoker participants, and analyses exploring multiple exposure to several types of sweeteners. All tests were 2-sided, and $P < 0.05$ was considered statistically significant. The statistical analysis software SAS, version 9.4, was used for analyses.

## Results

### Descriptive characteristics

In total, 102,865 participants (78.5% women) were included in the analyses (detailed flowchart shown in Fig 1). Average age at baseline was 42.2 ± 14.5 years. Average number of 24-hour dietary records per participant was 5.6 (SD = 3.0). Artificial sweeteners were consumed by 36.9% of the participants. Table 1 shows baseline characteristics of the study population by categories of total artificial sweetener intake. Compared to non-consumers (unadjusted descriptive comparisons), higher consumers tended to be more often women, younger, smokers, less physically active, more educated, and more likely to have prevalent diabetes. They had lower energy, alcohol, saturated fatty acid, fibre, fruit and vegetables, and whole-grain food intakes and higher intakes of sodium, total sugar, dairy products, sugary foods and drinks, and unsweetened non-alcoholic beverages. The main artificial sweetener was aspartame, contributing to 58% of intakes, followed by acesulfame-K (29%) and sucralose (10%) (Fig 2). These 3 sweeteners were respectively consumed by 28%, 34%, and 14% of the study population. All participants' intakes of aspartame and acesulfame-K were below the ADIs of 40 mg/kg body weight/day and 9 mg/kg body weight/day, respectively; only 5 participants exceeded the ADI of 15 mg/kg body weight/day for sucralose [5]. Soft drinks with no added sugars, table-top sweeteners, and yogurt/cottage cheese were the main contributors to total artificial sweetener intake, accounting for 53%, 29%, and 8% of intakes, respectively (Fig 3). Table A in S1 Appendix displays the number and percent of participants consuming 1, 2, or 3 of the main artificial sweeteners (aspartame, acesulfame-K, and sucralose). Participants frequently co-consumed several types of artificial sweeteners, but the proportion of those who were consumers of all 3 main artificial sweeteners was low (only 7.1%).

### Associations between intakes of artificial sweeteners and cancer risk

During follow-up (708,905 person-years, median follow-up time = 7.7 years, interquartile range = 4.7–9.4 years), 3,358 incident cancer cases were diagnosed (among which were 982

**Table 1. Baseline characteristics of the study population, NutriNet-Santé cohort, France, 2009–2021 ($n$ = 102,865)[1].**

| Characteristic | All participants | Categories of artificial sweetener intake[2] | | | P value[3] |
|---|---|---|---|---|---|
| | | Non-consumers | Lower consumers | Higher consumers | |
| Number of participants | 102,865 | 64,892 (63.08) | 18,986 (18.46) | 18,987 (18.46) | |
| Age (years), mean (SD) | 42.22 (14.50) | 42.82 (14.70) | 42.10 (14.54) | 40.31 (13.57) | <0.001 |
| Female sex | 80,711 (78.46) | 49,349 (76.05) | 15,681 (82.59) | 15,681 (82.59) | <0.001 |
| Height (cm), mean (SD) | 166.93 (8.18) | 167.24 (8.28) | 166.17 (7.94) | 166.61 (8.00) | <0.001 |
| Body mass index (kg/m$^2$), mean (SD) | 23.69 (4.48) | 23.29 (4.17) | 23.79 (4.49) | 24.96 (5.20) | <0.001 |
| Family history of cancer | 39,040 (37.95) | 26,643 (37.97) | 7,493 (39.46) | 6,904 (36.36) | <0.001 |
| Prevalent type 1 diabetes | 254 (0.25) | 118 (0.18) | 43 (0.23) | 93 (0.49) | <0.001 |
| Prevalent type 2 diabetes | 1,522 (1.48) | 676 (1.04) | 321 (1.69) | 525 (2.76) | <0.001 |
| Educational level | | | | | <0.001 |
| Less than high school degree | 18,062 (17.42) | 11,523 (17.75) | 3,263 (17.19) | 3,276 (17.25) | |
| 2 years or less after high school | 17,921 (17.42) | 11,269 (17.36) | 3,304 (17.40) | 3,3348 (17.63) | |
| More than 2 years after high school | 66,894 (65.02) | 41,109 (64.88) | 12,420 (65.41) | 12,365 (65.12) | |
| Smoking status | | | | | <0.001 |
| Current | 17,945 (17.44) | 11,188 (17.24) | 2,898 (15.26) | 3,859 (20.32) | |
| Former | 33,030 (32.11) | 20,576 (31.70) | 6,031 (31.76) | 6,423 (33.82) | |
| Never | 51,902 (50.45) | 33,137 (51.06) | 10,058 (52.97) | 8,707 (45.85) | |
| Physical activity level[4] | | | | | <0.001 |
| Low | 21,443 (20.84) | 13,159 (20.28) | 4,070 (21.44) | 4,214 (22.19) | |
| Moderate | 38,152 (37.09) | 23,910 (36.84) | 7,310 (38.50) | 6,932 (36.51) | |
| High | 29,023 (28.21) | 18,919 (29.15) | 5,093 (26.82) | 5,011 (26.39) | |
| Number of biological children, mean (SD) | 1.28 (1.28) | 1.32 (1.31) | 1.26 (1.23) | 1.18 (1.21) | <0.001 |
| Menopausal or peri-menopausal | 28,694 (35.54) | 18,019 (36.51) | 5,940 (37.88) | 4,735 (30.19) | <0.001 |
| Hormonal treatment for menopause[5] | 3,482 (4.31) | 2,064 (4.18) | 738 (4.71) | 680 (4.34) | 0.0187 |
| Oral contraception[6] | 22,991 (28.48) | 13,052 (26.44) | 4,740 (30.23) | 5,199 (33.15) | <0.001 |
| Energy intake without alcohol (kcal/day), mean (SD) | 1901.69 (471.70) | 1913.09 (478.76) | 1895.27 (435.87) | 1869.15 (480.16) | <0.001 |
| Alcohol intake (g/day), mean (SD) | 7.81 (11.88) | 8.12 (12.31) | 7.65 (11.09) | 6.89 (11.05) | <0.001 |
| Saturated fatty acid intake (g/day), mean (SD) | 33.21 (12.19) | 33.57 (12.34) | 33.22 (11.25) | 31.95 (12.46) | <0.001 |
| Sodium intake (mg/day), mean (SD) | 2719.72 (892.27) | 2709.80 (905.87) | 2728.75 (826.30) | 2744.62 (908.26) | <0.001 |
| Dietary fibre intake (g/day), mean (SD) | 19.48 (7.26) | 19.82 (7.56) | 19.03 (6.32) | 18.77 (7.02) | <0.001 |
| Total sugar intake (g/day), mean (SD) | 93.47 (33.45) | 92.93 (33.85) | 95.45 (31.03) | 93.35 (34.34) | <0.001 |
| Added sugar intake (g/day), mean (SD) | 38.58 (23.92) | 38.35 (23.73) | 40.12 (22.69) | 37.84 (25.66) | <0.001 |
| Percentage of energy from added sugar, mean (SD) | 7.95 (4.18) | 7.88 (4.15) | 8.31 (3.97) | 7.85 (4.45) | <0.001 |
| Sugary drink intake (ml/day), mean (SD) | 47.94 (107.32) | 42.81 (103.77) | 55.54 (99.11) | 57.90 (124.64) | <0.001 |
| Fruit and vegetable intake (g/day), mean (SD) | 405.11 (220.56) | 409.05 (223.10) | 399.24 (198.46) | 397.54 (232.19) | <0.001 |
| Whole-grain food intake (g/day), mean (SD) | 34.46 (46.52) | 36.01 (49.66) | 31.67 (38.98) | 31.98 (41.91) | <0.001 |
| Dairy product intake (g/day), mean (SD) | 196.48 (148.63) | 183.56 (145.11) | 202.70 (138.01) | 234.40 (163.17) | <0.001 |
| Ultra-processed food intake (percent of the diet in g/day), mean (SD) | 17.47 (9.98) | 16.04 (9.17) | 17.50 (8.73) | 22.32 (12.07) | <0.001 |
| Weight-loss diet during the first 2 years of follow-up | 17,569 (17.08) | 7,747 (11.94) | 3,626 (19.10) | 6,196 (32.63) | <0.001 |
| Total artificial sweetener intake (mg/day), mean (SD) | 16.07 (49.74) | 0.00 (0.00) | 7.62 (5.05) | 79.43 (91.72) | <0.001 |
| Aspartame (E951) intake (mg/day), mean (SD) | 9.35 (31.84) | 0.00 (0.00) | 3.24 (4.06) | 47.42 (60.75) | <0.001 |
| Acesulfame-K (E950) intake (mg/day), mean (SD) | 4.64 (15.14) | 0.00 (0.00) | 2.74 (2.86) | 22.39 (29.01) | <0.001 |
| Sucralose (E955) intake (mg/day), mean (SD) | 1.59 (16.21) | 0.00 (0.00) | 1.09 (1.98) | 7.52 (37.08) | <0.001 |

[1]Values are given as number (percentage) unless stated otherwise. 1 kcal = 4.18 kJ = 0.00418 MJ.

[2]Lower consumers and higher consumers were separated by the sex-specific median among consumers, i.e., 17.44 mg/day in men and 19.00 mg/day in women.

[3]P values for crude comparison between the 3 categories of sweetener intake by ANOVA or $\chi^2$ test as appropriate.

[4]Available for 88,618 participants, categorised into high, moderate, and low categories according to International Physical Activity Questionnaire guidelines.

[5]Among menopausal women.

[6]Among non-menopausal women.

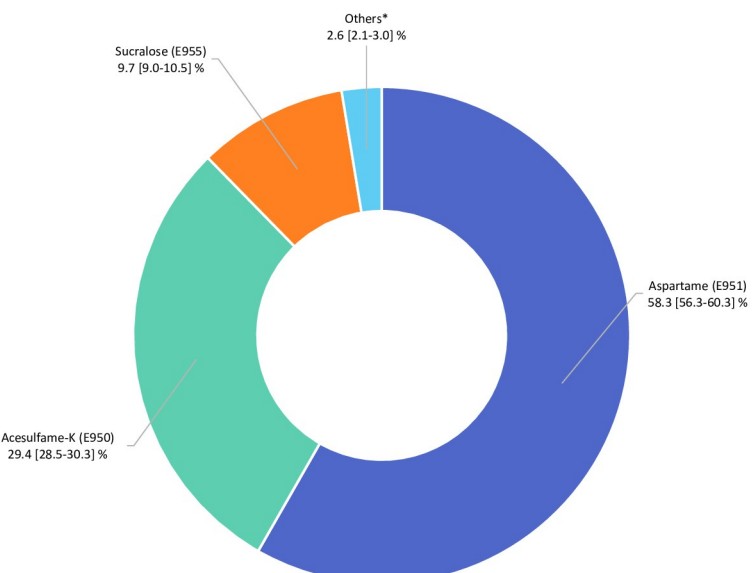

**Fig 2. Relative contribution of each specific artificial sweetener to the total intake of artificial sweeteners (percentage), NutriNet-Santé, France, 2009–2021 (*n* = 102,865).** *Cyclamates (E952), saccharin (E954), thaumatin (E957), neohesperidine dihydrochalcone (E959) steviol glycosides (E960), aspartame-acesulfame salt (E962).

breast, 403 prostate, and 2,023 obesity-related cancers). Average age at diagnosis was 59.5 ± 12.2 years.

Artificial sweetener intake was positively associated with the risk of overall cancer (HR for higher consumers versus non-consumers = 1.13 [95% CI 1.03 to 1.25], *P*-trend = 0.002)

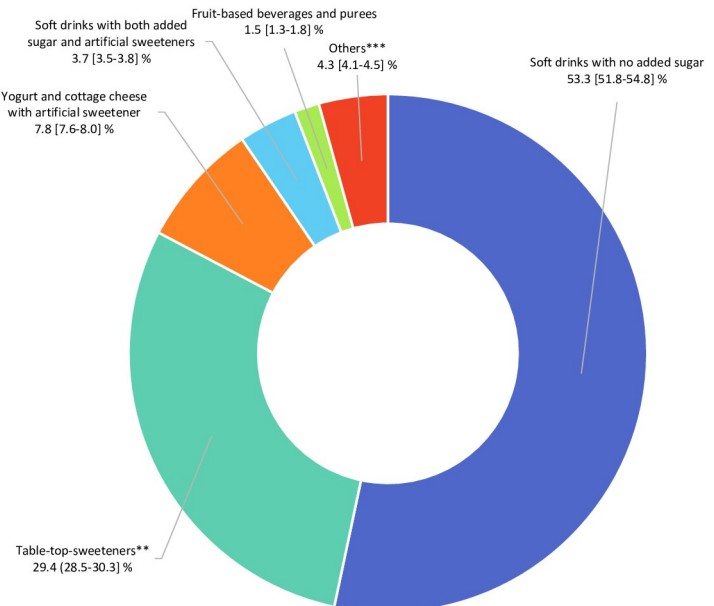

**Fig 3. Relative contribution of each food group to the total intake of artificial sweeteners (percentage), NutriNet-Santé, France, 2009–2021 (*n* = 102,865).** **Artificial sweeteners used as tablets, liquid, or powder, added by the participants in yogurts, hot drinks, etc., or for cooking. ***High-protein food substitutes, sugary foods, cookies, biscuits, cakes, pastries, breakfast cereals, sauces, savoury foods, and ultra-processed fish products.

(Table 2). In particular, higher cancer risks were observed for aspartame (HR = 1.15 [95% CI 1.03 to 1.28], $P$ = 0.002) and acesulfame-K (HR = 1.13 [95% CI 1.01 to 1.26], $P$ = 0.007). Increased risks were observed for breast cancer (HR = 1.22 [95% CI 1.01 to 1.48], $P$ = 0.036, for aspartame) and obesity-related cancers (HR = 1.13 [95% CI 1.00 to 1.28], $P$ = 0.036, for total artificial sweeteners, and HR = 1.15 [95% CI 1.01 to 1.32], $P$ = 0.026, for aspartame). Overall, the same direction of association was observed in pre- and postmenopausal women (Table B in S1 Appendix). Heterogeneity tests showed no difference between pre- and post-menopausal models for total artificial sweeteners, aspartame, and acesulfame-K ($P$ for heterogeneity = 0.440, 0.332, and 0.539, respectively). $P$ for heterogeneity was 0.015 for sucralose, but associations with cancer risk were non-significant in both strata for this food additive, with a low number of consumers per strata. No association was found with prostate cancer (Table 2). Forest plots in Fig B in S1 Appendix present both minimally and fully adjusted associations, showing similar results. Results for competing risk analyses are presented in Result A in S1 Appendix.

Associations in the non-consumer versus consumer analyses (Table C in S1 Appendix) were consistent with those in the non-consumer versus lower and higher consumer analyses, and the heterogeneity tests performed revealed no difference between the two-category model and the three-category model (all $P$ > 0.05). The comparison of the model with 3 categories of intake to the model with a linear trend across categories did not provide evidence of non-linearity ($P$ = 0.107, 0.250, 0.348, and 0.437 for total artificial sweeteners, aspartame, acesulfame-K, and sucralose, respectively, for the overall cancer model). After adjustment for the total dose of artificial sweetener exposure, cancer risk did not differ between participants consuming 1, 2, or 3 different sweeteners ($P$ > 0.05 for all 2-by-2 comparisons).

No interaction was detected for any cancer outcome between artificial sweetener exposures and BMI, nor between the 3 main artificial sweeteners (Table D in S1 Appendix).

We additionally investigated a 6-category composite variable, combining artificial sweetener and sugar intakes, which revealed increased cancer risk associated with both artificial sweetener and sugar intakes (Fig C and Table E in S1 Appendix). In particular, no difference was detected between the categories 'higher artificial sweetener consumption and sugar intake below the official recommended limit' and 'no artificial sweetener consumption and sugar intake exceeding the recommended limit' (Table F in S1 Appendix). Overall, results remained similar in all sensitivity analyses (Table G in S1 Appendix).

## Discussion

Results from this large-scale population-based cohort study suggest a positive association between higher intake of artificial sweeteners (especially aspartame and acesulfame-K) and overall cancer risk. More specifically, aspartame intake was associated with increased breast and obesity-related cancers.

To our knowledge no previous cohort study has directly investigated the association between quantitative artificial sweetener intakes per se—distinguishing the different types of sweeteners, in the whole diet—and cancer risk. However, some proxies have been used. Aspartame intake from ASBs was assessed in the NIH-AARP Diet and Health Study cohort [16], and no association with hematopoietic and brain cancers was revealed. Intakes through table-top sweeteners added by participants was additionally considered in 2 large-scale American cohorts (the Nurses' Health Study and the Health Professionals Follow-Up Study) [2] and in the Cancer Prevention Study II (CPS-II) Nutrition Cohort [39]. Results from these studies were conflicting; McCullough et al. found no associations with non-Hodgkin lymphoma in the CPS-II cohort [39]. In contrast, Schernhammer et al. [2], who adjusted their model for

**Table 2. Association between total artificial sweetener, aspartame, acesulfame-K, and sucralose intakes (mg/day) and cancer risk, NutriNet-Santé cohort, France, 2009–2021 (n = 102,865)[1].**

| Cancer site | Exposure (mg/day) | Measure | Non-consumers | Lower consumers[2] | Higher consumers[2] | P-trend |
|---|---|---|---|---|---|---|
| All cancers | Total artificial sweeteners | Participants/incident cases | 64,892/2,013 | 18,986/744 | 18,987/601 | |
| | | HR (95% CI)—minimally adjusted[3] | 1 | 1.26 (1.16 to 1.37) | 1.19 (1.08 to 1.30) | <0.001 |
| | | HR (95% CI)—fully adjusted[4] | 1 | 1.14 (1.05 to 1.25) | 1.13 (1.03 to 1.25) | 0.002 |
| | Aspartame | Participants/incident cases | 74,169/2,309 | 14,345/572 | 14,351/477 | |
| | | HR (95% CI)—minimally adjusted | 1 | 1.21 (1.11 to 1.33) | 1.18 (1.07 to 1.31) | <0.001 |
| | | HR (95% CI)—fully adjusted | 1 | 1.12 (1.02 to 1.23) | 1.15 (1.03 to 1.28) | 0.002 |
| | Acesulfame-K | Participants/incident cases | 67,662/2,096 | 17,601/766 | 17,602/496 | |
| | | HR (95% CI)—minimally adjusted | 1 | 1.22 (1.12 to 1.33) | 1.19 (1.07 to 1.33) | <0.001 |
| | | HR (95% CI)—fully adjusted | 1 | 1.12 (1.03 to 1.22) | 1.13 (1.01 to 1.26) | 0.007 |
| | Sucralose | Participants/incident cases | 88,867/2,883 | 7,005/288 | 6,993/187 | |
| | | HR (95% CI)—minimally adjusted | 1 | 1.20 (1.06 to 1.35) | 1.00 (0.86 to 1.17) | 0.177 |
| | | HR (95% CI)—fully adjusted | 1 | 1.03 (0.91 to 1.17) | 0.96 (0.82 to 1.12) | 0.823 |
| Breast cancer | Total artificial sweeteners | Participants/incident cases | 49,349/556 | 15,681/229 | 15,681/194 | |
| | | HR (95% CI)—minimally adjusted | 1 | 1.23 (1.06 to 1.44) | 1.16 (0.99 to 1.37) | 0.019 |
| | | HR (95% CI)—fully adjusted | 1 | 1.11 (0.95 to 1.30) | 1.16 (0.97 to 1.38) | 0.064 |
| | Aspartame | Participants/incident cases | 56,721/647 | 11,999/176 | 12,000/156 | |
| | | HR (95% CI)—minimally adjusted | 1 | 1.17 (0.99 to 1.39) | 1.18 (0.98 to 1.42) | 0.031 |
| | | HR (95% CI)—fully adjusted | 1 | 1.09 (0.92 to 1.29) | 1.22 (1.01 to 1.48) | 0.036 |
| | Acesulfame-K | Participants/incident cases | 51,712/581 | 14,578/232 | 14,579/166 | |
| | | HR (95% CI)—minimally adjusted | 1 | 1.20 (1.03 to 1.40) | 1.22 (1.00 to 1.49) | 0.014 |
| | | HR (95% CI)—fully adjusted | 1 | 1.11 (0.95 to 1.30) | 1.17 (0.96 to 1.43) | 0.086 |
| | Sucralose | Participants/incident cases | 69,189/826 | 5,772/93 | 5,750/60 | |
| | | HR (95% CI)—minimally adjusted | 1 | 1.23 (0.99 to 1.52) | 0.99 (0.76 to 1.30) | 0.438 |
| | | HR (95% CI)—fully adjusted | 1 | 1.04 (0.84 to 1.30) | 0.93 (0.71 to 1.22) | 0.786 |
| Prostate cancer | Total artificial sweeteners | Participants/incident cases | 15,543/282 | 3,305/63 | 3,306/58 | |
| | | HR (95% CI)—minimally adjusted | 1 | 1.02 (0.78 to 1.34) | 1.20 (0.90 to 1.59) | 0.257 |
| | | HR (95% CI)—fully adjusted | 1 | 0.92 (0.70 to 1.22) | 1.26 (0.94 to 1.68) | 0.274 |
| | Aspartame | Participants/incident cases | 17,457/310 | 2,346/49 | 2,351/44 | |
| | | HR (95% CI)—minimally adjusted | 1 | 1.04 (0.77 to 1.41) | 1.19 (0.86 to 1.64) | 0.324 |
| | | HR (95% CI)—fully adjusted | 1 | 0.95 (0.70 to 1.30) | 1.28 (0.91 to 1.79) | 0.280 |
| | Acesulfame-K | Participants/incident cases | 16,108/288 | 3,023/76 | 3,023/39 | |
| | | HR (95% CI)—minimally adjusted | 1 | 1.13 (0.87 to 1.48) | 1.25 (0.86 to 1.80) | 0.184 |
| | | HR (95% CI)—fully adjusted | 1 | 1.06 (0.81 to 1.39) | 1.18 (0.82 to 1.71) | 0.365 |
| | Sucralose | Participants/incident cases | 19,678/365 | 1,233/25 | 1,243/13 | |
| | | HR (95% CI)—minimally adjusted | 1 | 1.02 (0.68 to 1.54) | 0.99 (0.57 to 1.74) | 0.967 |
| | | HR (95% CI)—fully adjusted | 1 | 0.86 (0.57 to 1.30) | 1.01 (0.57 to 1.77) | 0.699 |

(*Continued*)

**Table 2.** (Continued)

| Cancer site | Exposure (mg/day) | Measure | Non-consumers | Lower consumers[2] | Higher consumers[2] | P-trend |
|---|---|---|---|---|---|---|
| Obesity-related cancers | Total artificial sweeteners | Participants/incident cases | 64,892/1,232 | 18,986/433 | 18,987/358 | |
| | | HR (95% CI)—minimally adjusted | 1 | 1.20 (1.08 to 1.34) | 1.17 (1.04 to 1.32) | 0.001 |
| | | HR (95% CI)—fully adjusted | 1 | 1.08 (0.97 to 1.21) | 1.13 (1.00 to 1.28) | 0.036 |
| | Aspartame | Participants/Incident cases | 74,169/1,401 | 14,345/337 | 14,351/285 | |
| | | HR (95% CI)—minimally adjusted | 1 | 1.17 (1.04 to 1.31) | 1.17 (1.03 to 1.33) | 0.003 |
| | | HR (95% CI)—fully adjusted | 1 | 1.08 (0.96 to 1.22) | 1.15 (1.01 to 1.32) | 0.026 |
| | Acesulfame-K | Participants/Incident cases | 67,662/1,275 | 17,601/457 | 17,602/291 | |
| | | HR (95% CI)—minimally adjusted | 1 | 1.18 (1.05 to 1.31) | 1.17 (1.02 to 1.35) | 0.004 |
| | | HR (95% CI)—fully adjusted | 1 | 1.09 (0.97 to 1.22) | 1.13 (0.97 to 1.30) | 0.064 |
| | Sucralose | Participants/Incident cases | 88,867/1,756 | 7,005/167 | 6,993/100 | |
| | | HR (95% CI)—minimally adjusted | 1 | 1.14 (0.97 to 1.33) | 0.90 (0.73 to 1.11) | 0.899 |
| | | HR (95% CI)—fully adjusted | 1 | 0.98 (0.84 to 1.16) | 0.87 (0.71 to 1.07) | 0.230 |

[1]Median follow-up times for all, breast, prostate, and obesity-related cancers were, respectively, 7.7, 7.6, 8.0, and 7.7 years. Person-years were, respectively, 708,905, 551,803, 157,102, and 708,905.

[2]The sex-specific cutoffs between higher and lower consumers were 17.44 mg/day in men and 19.00 mg/day in women for total artificial sweeteners, 14.45 mg/day in men and 15.39 mg/day in women for aspartame, 5.06 mg/day in men and 5.50 mg/day in women for acesulfame-K, and 3.46 mg/day in men and 3.43 mg/day in women for sucralose.

[3]Minimally adjusted models were adjusted for age (time scale) and sex (except for breast and prostate cancer).

[4] Fully adjusted multivariable Cox proportional hazards models (main model) were adjusted for age (time scale), sex (except for breast and prostate cancer), BMI (continuous, $kg/m^2$), height (continuous, cm), percentage weight gain during follow-up (continuous), physical activity (categorical International Physical Activity Questionnaire variable: high, moderate, low, missing value), smoking status (categorical: never, former, current), number of smoked cigarettes in pack-years (continuous), educational level (categorical: less than high school degree, ≤2 years after high school degree, >2 years after high school degree), number of 24-hour dietary records (continuous), family history of cancer (categorical: yes, no), prevalent diabetes (categorical: yes, no), energy intake without alcohol (continuous variable: kcal/day), and daily intakes (continuous, g/day) of alcohol, sodium, saturated fatty acids, fibre, sugar, fruit and vegetables, whole-grain foods, and dairy products. Breast cancer models were also adjusted for age at menarche (categorical: <12 years old, ≥12 years old), age at first child (categorical: no child, <30 years, ≥30 years), number of biological children (continuous), baseline menopausal status (categorical: menopausal, non-menopausal), oral contraceptive use at baseline and during follow-up (categorical: yes, no), and hormonal treatment for menopause at baseline and during follow-up (categorical: yes, no). In addition, all models were mutually adjusted for artificial sweetener intake other than the one studied.

various additional dietary factors, found increased risks among male participants for non-Hodgkin lymphoma and multiple myeloma. Other studies did not investigate artificial sweeteners but the whole group of ASBs in millilitres or servings per day. A recent systematic review and meta-analysis of sweetened beverages and risk of cancer at different sites [40] stressed the lack of studies on ASBs and cancer risk except for pancreatic cancer, for which they found a positive, although non-significant, association. Likewise, previous analyses of the NutriNet-Santé cohort did not detect an association between ASBs and cancer risk [41], suggesting that measurements of ASBs might be inadequate to accurately characterise the overall dietary exposure to artificial sweeteners. However, ASBs have recently been investigated within the Melbourne Collaborative Cohort Study, revealing a positive association with cancers not related to obesity [42] but not with obesity-related cancers [43]. Evaluating sweetener intake through ASBs might not be sufficient since many other foods are also vectors of artificial sweeteners (e.g., breakfast cereals, yogurts, ice creams, and table-top sweeteners). Several case–control studies have analysed associations between artificial sweeteners or ASBs and different cancer locations, as recently meta-analysed [44–46]. Although these studies bring interesting pieces of evidence, potentially strong reverse causality bias with this type of design limits the interpretability of these studies. It is therefore more appropriate to rely on large-scale prospective

studies when available. Several randomized control trials have tested the effect of artificial sweeteners on health parameters such as body weight, BMI, glycaemic control, and eating behaviour [7]. But none, to our knowledge, has considered cancer as a primary or secondary outcome. In a previous publication, we showed that sugar intake was also associated with increased overall (HR for quartile 4 versus quartile 1 = 1.17 [95% CI 1.00 to 1.37], $P = 0.02$) and breast (HR for quartile 4 versus quartile 1 = 1.51 [95% CI 1.14 to 2.00], $P = 0.001$) cancers in this cohort [47]. In the present study, the fact that no difference was detected between the categories 'higher artificial sweetener consumption and sugar intake below the official recommended limit' and 'no artificial sweetener consumption and sugar intake exceeding the recommended limit' suggests that artificial sweeteners and excessive sugar intake may be equally associated with cancer risk.

On the one hand, obesity is a recognised risk factor for many cancers [35]. On the other hand, although it remains unclear, associations between artificial sweetener intake and weight gain have been suggested [8,10,48–51]. Thus, we investigated the associations between artificial sweetener intake and the risk of obesity-related cancers. The positive associations observed suggest that part of this relationship may be driven by overweight-related metabolic disturbances. However, the associations between artificial sweetener intake and cancer risk observed in this study are not entirely explained by weight-gain-related mechanisms, since the models were adjusted for baseline BMI and weight gain during follow-up. Other mechanisms could be involved. Carcinogenicity of artificial sweeteners has long been suspected based on in vitro and in vivo experimental results. Although results from animal studies remain controversial [52–54], some results obtained in rodent models suggested that aspartame was associated with higher risks of different cancers (lymphomas and leukaemias and hepatocellular and alveolar/bronchiolar carcinomas) [11] at dose levels comparable to those to which humans can be exposed. Although these findings have been controversial [55], additional data have been recently published supporting the original findings from the Ramazzini Institute regarding the identification of tumours [56]. This suggests the need for an updated evaluation of aspartame's carcinogenicity and ADI. Aspartame's toxicity has also been investigated in several in vitro studies [14,15], the results of which suggested its carcinogenicity [14], potentially through mechanisms related to inflammation, angiogenesis, promotion of DNA damage, and inhibition of apoptosis [15]. More recently, sucralose was shown to increase the risk of malignant tumours and hematopoietic neoplasia in mice [17]. An in vivo study found that acesulfame-K and saccharin elicited even more DNA damage than aspartame [13]. Lastly, Suez et al. revealed findings implicating non-caloric artificial sweeteners (saccharin, sucralose, and aspartame) in the modification of gut microbiota (induction of dysbiosis and glucose intolerance in mice and in healthy humans) [57], which in turn might be involved in the development of some cancers [58].

Beyond its longitudinal design and large sample size, one major strength of our study is its detailed assessment of exposure to artificial sweeteners at the individual level [30]. The repeated 24-hour dietary records allowed us to collect precise information on the consumption of industrial products, including their commercial brands/names. These consumption data were merged with data from several large composition databases and results from thousands of laboratory assays in food matrixes. Also, dynamic matching with the date of consumption was used to select the most appropriate composition data, which reduced potential bias due to reformulations. Thus, total artificial sweetener intake through sources other than just ASBs could be considered.

However, several limitations should be acknowledged. First, caution must be taken in extrapolating the results to the whole adult French population. As generally observed in volunteer-based cohorts, participants were more likely to be women, to have higher educational and socio-professional levels, to have health-conscious behaviours (diet and lifestyle) [59], and to

be older on average (while artificial sweetener intake is lower in older individuals). This could contribute to explaining the relatively lower intake levels of artificial sweeteners compared to those described in the literature for national studies (e.g., for aspartame, 0.53/0.40 for women/ men in NutriNet-Santé, compared to 0.81/1.08 estimated in simulations for the French population) [4,60]. This suggests that the associations with cancer risk observed among NutriNet-Santé participants may underestimate what would be observed in the general population with a broader range of exposure. In particular, the absence of a relationship between sucralose and cancer risk in this study should be considered with caution since exposure to sucralose was very low compared to the exposures for aspartame and acesulfame-K. However, differences in exposure estimates may also be due to more precise assessment at the individual level in the present study than in simulation studies based on average information for generic product categories. Second, the limited number of cases prevented us from studying associations for other cancer sites (e.g., pancreatic, ovarian, endometrial, kidney, liver, and bladder) than the main ones presented here. Lastly, causal links cannot be established by this unique study; in particular, residual confounding bias cannot be entirely ruled out, although the wide range of adjustment factors accounted for in main and sensitivity analysis models limited this risk. To assess the causal association between cancer incidence and the intake of artificial sweeteners and sugar, genetic markers linked with sweet taste preference (e.g., the rs838133 variant of hepatokine fibroblast growth factor 21 [61]) could be integrated into a polygenic score that could be used as part of a Mendelian randomization study.

This large-scale population-based cohort study suggests associations between artificial sweeteners, especially aspartame and acesulfame-K, and cancer risk, more specifically breast and obesity-related cancers. These results need to be replicated in other large-scale cohorts, and underlying mechanisms clarified by experimental studies. Artificial sweeteners are present in many food and beverage brands worldwide [3] and are consumed by millions of citizens and patients daily. Our findings do not support the use of artificial sweeteners as safe alternatives for sugar in foods or beverages and provide important and novel information to address the controversies about their potential adverse health effects. These results are particularly relevant in the context of the ongoing in-depth re-evaluation of artificial sweeteners by EFSA and other agencies globally.

## Supporting information

**S1 STROBE-nut Checklist.**
(DOCX)

**S1 Appendix. Supplementary material.** Method A: Methodology for identification of energy under-reporting and validation studies for the 24-hour web-based dietary records. Fig A: Proportional hazards assumption testing using rescaled Schoenfeld-type residuals for the association between total artificial sweetener intake and cancer risk, NutriNet-Santé cohort, France, 2009–2021 ($n$ = 102,865). Method B: Multiple imputation by chained equations. Table A: Number of participants in each combination of artificial sweetener consumption for aspartame, acesulfame-K, and sucralose (non-consumers, consumers of 1 type of artificial sweetener, consumers of 2 types, and consumers of 3 types), NutriNet-Santé cohort, France, 2009–2021 ($n$ = 102,865). Table B: Associations between total artificial sweetener, acesulfame-K, aspartame, and sucralose intakes (mg/day) and breast cancer risk in premenopausal and postmenopausal women, NutriNet-Santé cohort, France, 2009–2021 ($n$ = 80,711). Fig B: Forest plots presenting the minimally adjusted and fully adjusted associations between total artificial sweetener, aspartame, acesulfame-K, and sucralose intakes (mg/day) and cancer risk, NutriNet-Santé cohort, France, 2009–2021 ($n$ = 102,865). Result A: Results from competing risk

analyses (cause-specific Cox proportional hazards models). Table C: Association between total artificial sweetener, aspartame, acesulfame-K, and sucralose intakes (mg/day) and cancer risk, NutriNet-Santé cohort, France, 2009–2021 ($n$ = 102,865)—consumers versus non-consumers. Table D: Interaction tests for BMI and between the 3 main artificial sweeteners (aspartame, acesulfame-K, and sucralose) for all studied outcomes. Fig C: Cancer risk associated with combined artificial sweetener and sugar intakes, NutriNet-Santé cohort, France, 2009–2021 ($n$ = 102,865). Table E: Overall cancer risk associated with combined artificial sweetener and sugar intakes: 2-by-2 comparisons across categories, NutriNet-Santé cohort, France, 2009–2021 ($n$ = 102,865). Table F: Focus on the comparisons of cancer risk for participants with higher artificial sweetener consumption/lower sugar intake versus participants with no artificial sweetener consumption/higher sugar intake, NutriNet-Santé cohort, France, 2009–2021 ($n$ = 102,865). Table G: Association between artificial sweetener intake (mg/day) and cancer risk, NutriNet-Santé cohort, France, 2009–2021 ($n$ = 102,865)—sensitivity analyses. (DOCX)

## Acknowledgments

We thank Thi Hong Van Duong, Régis Gatibelza, Jagatjit Mohinder, and Aladi Timera (computer scientists); Julien Allegre, Nathalie Arnault, Laurent Bourhis, and Nicolas Dechamp (data manager/statisticians); Sandrine Kamdem (health event validator); and Maria Gomes (Nutrinaute support) for their technical contribution to the NutriNet-Santé study and Professor Raphaël Porcher for his help and expertise in biostatistics.

We thank all the volunteers of the NutriNet-Santé cohort.

## Author Contributions

**Conceptualization:** Charlotte Debras, Mathilde Touvier.

**Data curation:** Charlotte Debras.

**Formal analysis:** Charlotte Debras.

**Funding acquisition:** Mathilde Touvier.

**Investigation:** Charlotte Debras.

**Methodology:** Eloi Chazelas, Bernard Srour, Inge Huybrechts, Mathilde Touvier.

**Project administration:** Nathalie Druesne-Pecollo, Mathilde Touvier.

**Resources:** Eloi Chazelas, Fabien Szabo de Edelenyi, Cédric Agaësse, Alexandre De Sa, Rebecca Lutchia, Stéphane Gigandet.

**Software:** Eloi Chazelas, Younes Esseddik.

**Supervision:** Mathilde Touvier.

**Validation:** Eloi Chazelas, Bernard Srour, Inge Huybrechts, Chantal Julia, Emmanuelle Kesse-Guyot, Benjamin Allès, Valentina A. Andreeva, Pilar Galan, Serge Hercberg, Mélanie Deschasaux-Tanguy, Mathilde Touvier.

**Writing – original draft:** Charlotte Debras.

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
