## [Editor Report · Decision Letter 0]

11 Nov 2021

Dear Dr Debras, 

Thank you for submitting your manuscript entitled "Artificial sweeteners and cancer risk in the prospective NutriNet-Santé cohort" for consideration by PLOS Medicine.

Your manuscript has now been evaluated by the PLOS Medicine editorial staff and I am writing to let you know that we would like to send your submission out for external peer review.

Please re-submit your manuscript within two working days, i.e. by Nov 15 2021 11:59PM.

Kind regards,

Callam Davidson

Associate Editor

PLOS Medicine

---

## [Decision Letter · Decision Letter 1]

16 Dec 2021

Dear Dr. Debras,

Thank you very much for submitting your manuscript "Artificial sweeteners and cancer risk in the prospective NutriNet-Santé cohort" (PMEDICINE-D-21-04643R1) for consideration at PLOS Medicine. 

Your paper was evaluated by an associate editor and discussed among all the editors here. It was also discussed with an academic editor with relevant expertise, and sent to independent reviewers, including a statistical reviewer. The reviews are appended at the bottom of this email and any accompanying reviewer attachments can be seen via the link below:

[LINK]

In light of these reviews, I am afraid that we will not be able to accept the manuscript for publication in the journal in its current form, but we would like to consider a revised version that addresses the reviewers' and editors' comments. Obviously we cannot make any decision about publication until we have seen the revised manuscript and your response, and we plan to seek re-review by one or more of the reviewers. 

We hope to receive your revised manuscript by Jan 06 2022 11:59PM. Please email us (plosmedicine@plos.org) if you have any questions or concerns.

We look forward to receiving your revised manuscript. 

Sincerely,

Callam Davidson, 

PLOS Medicine

plosmedicine.org

Comments from the academic editor:

‘The dietary assessment method for this study is unique and strong - better than most other cohort studies. However, I have some concerns about potential residual confounding effects. The authors need to carefully address this issue in the revision’

Please revise your title according to PLOS Medicine's style. Your title must be nondeclarative and not a question. It should begin with main concept if possible. "Effect of" should be used only if causality can be inferred, i.e., for an RCT. Please place the study design ("A cohort study") in the subtitle (ie, after a colon).

* Please structure your abstract using the PLOS Medicine headings (Background, Methods and Findings, Conclusions).

* Please combine the Methods and Findings sections into one section, “Methods and findings”.

Abstract Methods and Findings:

* Please include the study design and length of follow up.

* Please include the important dependent variables that are adjusted for in the analyses.

Citations should be in square brackets, and preceding punctuation.

Supplementary Figure 1 is central to the understanding of the paper. Please incorporate it into the main paper.

Please remove references to results tables from the methods (lines 177, 192, 205, 215, and 227).

Please ensure that the study is reported according to the STROBE guideline, and include the completed STROBE checklist as Supporting Information. Please add the following statement, or similar, to the Methods: "This study is reported as per the Strengthening the Reporting of Observational Studies in Epidemiology (STROBE) guideline (S1 Checklist)."

Did your study have a prospective protocol or analysis plan? Please state this (either way) early in the Methods section.

In Figure 3, please ensure that the y axis is identical for all panels to facilitate comparison.

Throughout, please report P values to 3 decimal places (anything smaller please report as P < 0.001).

Please remove subheadings from your Discussion.

Please begin the Discussion with a short, clear summary of the article's findings.

Please remove the Competing interest statement, Contributorship statement and guarantor, Transparency statement, Copyright/license for publication statement (all PLOS content is published under a CC-BY license), Data sharing statement, Funding statement and statement of independence of researchers from funders, and LARC disclaimer from the main text. If published, all of this information will be captured as metadata based on your responses to the submission form. 

Please relocate the Patient involvement statement to the Methods section.

Comments from the reviewers:

Reviewer #1: The Nutri Net-Santé study provides rare data with which to examine the association between artificial sweeteners and health outcomes. It is also a very large data base as required for such work.

Line 29, 'contrasted' seems the wrong word here.

Line 49, remove 'references'.

Line 58, not well written, maybe '…liking for sweet taste is widespread globally…' 

Line 65, 360 why do you have 'references' after food? Would 'food products' be correct?

Line 68, 'saccharose' is same as sucrose but I would have thought sucrose was better known so would use this term.

In Supplementary Figure 2 'Schoenfeld' is spelt correctly in the heading but not within the figure.

Reviewer #2: Alex McConnachie, Statistical Review

The paper by Debras et al presents analyses of a large, prospective cohort study, looking at the association between dietary intake of artificial sweeteners and the incidence of cancers. This review considers the statistical parts of the paper.

In short, the statistics in this paper are very good. The background and data sources are well described. The exclusions due to apparent under-reporting are justified. The exclusion of early events to protect against reverse causation is good. Cox models are appropriate, and the key proportional hazards assumption is checked. Models are fully adjusted for a range of potential confounders, and multiple imputation is used to account for missing baseline data. Competing risks models are considered as one of several sensitivity analyses. The results are clearly presented.

The few comments that I do have are fairly minor.

The authors use cubic splines to check linearity. This would be fine, if the main analysis used consumption as a continuous variable, but that is not the case. Each exposure is modelled as a 3-level categorical term (no intake, low intake, high intake), and the significance is tested as a linear trend across categories. It is probably possible for there to exist a linear association between risk and intake as a continuous variable, but a non-linear association between risk and the three categories of intake. There are a few options that I can see: the authors could remove the linearity check from the paper; they could present the HR for each intake as a linear variable (possibly just in the supplements), in which case the linearity check would be relevant; or they could assess linearity in relation to the trend across categories, by comparing the model with 3 categories of intake to a model with a linear trend across categories.

The adjustments in the models are comprehensive, which is good. In similar papers that I have read, it is common to report the unadjusted association, followed by the associations after increasing levels of adjustment for groups of potential confounders. Forest plots can be a good way to present these associations. I would not consider this essential, but it is sometimes interesting to see the size of the unadjusted association, and the extent to which it is confounded by other things, and then which types of factors contribute most to this confounding. Whilst unlikely, it could even be the case that without adjustment, artificial sweetener intake is not associated with outcomes, and the association is only apparent after adjustment.

Line 263-265 states "Analyses conducted among pre- and postmenopausal women distinctively showed that the associations with breast cancers tended to be more pronounced in postmenopausal women." Looking at table S3, I am not convinced. Taking total intakes, the estimated HRs in premenopausal women are 1.21 and 1.24; in postmenopausal women, they are 1.19 and 1.30 - i.e. they are quite similar. The fact that these associations are statistically significant in postmenopausal women only is neither here nor there - I should think that there is no real evidence that the HRs are any different. Any conclusion about different levels of association should be based on statistical interaction tests.

Line 273, the HR for sugar intake is reported as "1." - I reckon this should be about 1.17. Is this in a model that accounts for artificial sweetener intake? This finding is interesting - does it mean that the risk of cancer is roughly the same if someone replaces their sugar intake with artificial sweeteners? How does that affect the conclusion that artificial sweeteners are not a safe alternative to sugar? If the risk of cancer is about the same, but the risk of metabolic conditions is reduced (if that is the case), then maybe they are a safer alternative?

Obviously not for this paper, but are there genetic markers for how much of a "sweet tooth" someone has? Do these factors predict both sugar and artificial sweetener intake? Could a polygenic score be used as part of a Mendelian Randomisation study, to assess the causal association between cancer incidence and the intake of artificial sweeteners (and sugar)?

Reviewer #3: Artificial sweeteners and cancer risk in the prospective NutriNet-Sante cohort

In this paper, the authors examined the association of intake of artificial sweeteners and risk of cancer (both overall and site specific) in approximately 100k participants from France. They find that intake of artificial sweeteners is positively associated with risk of cancer, especially breast cancer. Aspartame and acesulfame-K were each individually associated with increased risk of cancer.

This study is clearly scientifically important. Artificial sweeteners, including aspartame and acesulfame-K, are present in thousands of foods and consumed by millions of people. Any association with cancer risk would be scientifically important. The study also addresses a clear gap in the literature. Few epidemiologic studies have examined the association of artificial sweeteners with risk of cancer, particularly breast cancer. Few cohorts have even meaningfully measured or quantified artificial sweetener intake. 

A major strength of the current study is that the authors had previously quantified levels of artificial sweeteners in commonly consumed foods and beverages in France. This allows the present study to link its dietary measures to a food database with detailed measures of artificial sweetener composition, providing a superior and more granular measure of artificial sweetener intake than other cohorts have. The primary limitations are that this is only one study of one population and the sample size is not especially large for cancer epidemiology (which is why only a few specific types of cancer could be examined). Studies in cohorts like EPIC or the AARP or in research consortia often have many hundreds of thousands or millions of participants. With additional studies, it remains possible that these associations will converge toward the null in the aggregate, such as occurs with "Winner's Curse". To the authors' credit, they appropriately acknowledge their study's limitations. My recommendations for revisions are described below.

Specific comments

Abstract background: Artificial sweeteners were originally marketed for weight loss, not for prevention of chronic disease (and excess weight, in of itself, is not a chronic disease). In fact, sodas like Tab, introduced in 1963, pre-date much of the epidemiological evidence on sugar and chronic disease. Consequently, the cause-and-effect sequence described in the first sentence seems somewhat inaccurate.

Abstract background: "findings remain contrasted" should be rewritten as "findings conflict"

Introduction: Some parts of this text need light editing to conform with typical English usage, e.g. "even though these" should be replaced with "that", "expertises" should be "evaluations".

Cancer case ascertainment: Are there any methodological publications documenting the sensitivity and specificity of case ascertainment for the cohort? What percent of cancer cases are ascertained given the relatively recent date of January 22nd, 2021? Could there be a lag in reporting for some cases? If a publication is available, please include a citation here.

Statistical analysis: Please describe the rationale for reverse causation. Is the idea that some participants get sick with subclinical cancer, change their dietary habits, and then are diagnosed? Or is the concern perhaps more accurately described as recall bias?

Statistical analysis: Given that the first two years of cases are excluded, the authors should also state that the first two years of follow-up were also excluded to prevent "immortality bias", since no cancers could occur during this time (i.e. follow-up starts at age at entry into cohort plus two years). 

Results, Supplementary Figure 3: The shaded areas do not appear proportionate to the percentages. If they are not proportionate, consider redrawing or presenting as a table. 

Results. Associations: Some of the results are hard to read, e.g. breast cancer (HR=1.25 (1.02 to 1.53) P=0.01, HR=1.33 (1.05 to 1.69) P=0.007…respectively). Consider changing use of punctuation to simplify readability.

Results, Associations: "Associations were consistent in the non-consumers vs. consumers (Supplementary Table 1)". This should include a formal test for heterogeneity.

Results, Associations: Was there any consideration given to presenting results for artificially sweetened beverages (ASB)? The results of the current study, at a superficial level, seem to differ from those of the prior BMJ study from this cohort. It would be interesting to know if results differ because of improved measurement in the current study or if measurements of ASB are simply inadequate to characterize artificial sweetener exposure. 

Table 1: please clarify what education "no" means. And the other education categories as well. Change "Education level" to "Education level, %". Modify other headings as needed to match.

Table 2: "(mg/d)" is not needed in this table except in the footnote.

Discussion, Mechanistic Plausibility: Reference 12 is a commentary and should not be cited as if it were an independent scientific study. Please also soften the tone of the phrase "recently confirmed in rats". The word "confirmed" implies multiple independent replications, but this is not what was done here. These additional data support, but do not confirm, the original findings.

References: Please update ref. 29 now that it is published.

Reviewer #4: Manuscript: Artificial sweeteners and cancer risk in the prospective NutriNet-Santé cohort

Manuscript # PMEDICINE-D-21-04643R1

General Comments

This manuscript examined the association of total and specific types of artificial sweetener consumption and cancer risk in a large web-based prospective cohort. Higher intake was associated with higher overall cancer, as well as breast and obesity-related cancers. This manuscript is well-written and contributes to the limited evidence on artificial sweeteners and cancer risk. Only a few minor comments are listed below.

Specific Comments

1. Results: Although the authors state that the number of high artificial sweetener consumers who consumed all three main types was low and that no interaction was detected between the three types, there are a notable number of participants who consumed at least 2 different types of sweetener. Food products also frequently contain more than one type of artificial sweetener. Can the authors explain whether they tried to investigate this further? This does not necessarily need to be added to the manuscript, but it would be interesting to understand whether the authors considered this since many individuals consumed more than one type of artificial sweetener.

2. Supplementary Table 2: It is commendable that the authors examined the intake of artificial sweeteners and sugar, as these may or may not be consumed together. Did the authors also compare additional groups, such as "No artificial sweetener and no sugar" and "High artificial sweetener and high sugar," etc. with the two groups included in this table?

3. Discussion, line 292 and 293: It appears that the abbreviation for Cancer Prevention Study-II should be CPS-II.

4. Discussion: It seems that the reason the authors examined the association of artificial sweeteners and obesity-related cancers was because of a potential link between artificial sweeteners and obesity. The manuscript would be strengthened with additional discussion about this.

[LINK]

---

## [Decision Letter · Decision Letter 2]

21 Jan 2022

Dear Dr. Debras,

Thank you very much for submitting your revised manuscript "Artificial sweeteners and cancer risk: results from the prospective NutriNet-Santé cohort" (PMEDICINE-D-21-04643R2) for consideration at PLOS Medicine. 

Your paper was evaluated by an associate editor and discussed among all the editors here. It was also discussed with an academic editor with relevant expertise, and sent back to independent reviewers, including a statistical reviewer. The reviews are appended at the bottom of this email and any accompanying reviewer attachments can be seen via the link below:

[LINK]

In light of these reviews, I am afraid that we will not be able to accept the manuscript for publication in the journal in its current form, but we would like to consider a revised version that addresses the reviewers' and editors' comments. We cannot make any decision about publication until we have seen the revised manuscript and your response, and we plan to seek re-review by one or more of the reviewers. 

We hope to receive your revised manuscript by Feb 11 2022 11:59PM. Please email us (plosmedicine@plos.org) if you have any questions or concerns.

We look forward to receiving your revised manuscript. 

Sincerely,

Callam Davidson, 

Associate Editor

PLOS Medicine

plosmedicine.org

Please update your title to ‘Artificial sweeteners and cancer risk: a population-based cohort study’ or similar (please do not include the word ‘prospective’).

Lines 49-51: Please shorten the limitations section of your abstract to a single sentence (e.g. ‘Limitations of this study include potential selection bias, residual confounding, and reverse causality, though sensitivity analyses were performed to address these concerns.’ or similar).

Lines 57-8: Please remove the trial registration number/URL from the abstract.

Line 71: Please quantify the key findings mentioned in your Author Summary (with 95% confidence intervals and p-values). 

Thank you for including the STROBE-nut checklist – please update the checklist to use section names/paragraph numbers as opposed to page numbers (which are liable to change during the revision process).

As no pre-specified analysis plan is available for this study, the term ‘prospective’ should be avoided throughout; please refer to the study instead as a population-based cohort study. If the authors wish to maintain the ‘prospective’ descriptor, it would be best if a document detailing the pre-specified analysis plan could be provided in the supporting information (the URL provided on lines 145-6 appears to be a news article). 

Similar to the above, apologies if I have missed it but I could not locate the general protocol at the provided URL (https://info.etude-nutrinet-sante.fr/siteinfo/), would it be possible to provide a direct link to the protocol?

Please confirm whether SG’s affiliation with Open Food Facts ought to be declared as a COI.

Lines 262/332: Please tabulate this data and present it in the supporting information.

Line 433: Please remove this subheading in the Discussion.

Please show the unadjusted comparisons as well as the adjusted comparisons in Table 2.

Comments from the reviewers:

Reviewer #2: Alex McConnachie, Statistical Review

I thank the authors for their consideration of my original points. I feel that the paper has been improved, though I have one remaining comment.

The data presented in supplementary table 2 do not seem to be in line with the interaction p-values reported. For total artificial sweeteners, the interaction p-value is given as 0.044, suggesting different associations between intake and premenopausal breast cancer compared to postmenopausal breast cancer. However, the hazard ratios do not appear to be very different. For low consumers vs. non-consumers, the HRs are the same (1.09), and for high consumers vs. non-consumers, the difference is small (1.15 vs. 1.20) with confidence intervals that entirely overlap. These HRs and CIs do not, in my opinion, seem consistent with an interaction p-value of 0.044. If anything, it is for sucralose, where the associations trend in opposite directions, where I might anticipate a significant interaction, but this interaction is reported as p=0.152.

Also, I spotted a couple of typos:

In supplementary table 3, for all cancers, sucralose, the HR is reported as 1.10, which is very close to the upper confidence limit. Should this be 1.01?

In supplementary table 4, the HR between [ No AS / Low sugar ] and [ Low AS / Low sugar ] should be less than 1, but is reported as 1.15, the same as the opposite comparison.

Reviewer #3: I am satisfied with the revisions made to this article.

Reviewer #4: The authors have adequately addressed the comments of the Editors and Reviewers.

Minor editing is needed to correct small errors (e.g., Introduction, lines 82-83- could be edited to "…reduce added sugar content and corresponding calories while maintaining sweetness"; Results, page 9, line 326- comma instead of semicolon for 0.250; Discussion, line 357- "CPS-II" cohort, etc.).

[LINK]

---

## [Decision Letter · Decision Letter 3]

17 Feb 2022

Dear Dr. Debras,

Thank you very much for re-submitting your manuscript "Artificial sweeteners and cancer risk: results from the NutriNet-Santé population-based cohort study" (PMEDICINE-D-21-04643R3) for review by PLOS Medicine.

I have discussed the paper with my colleagues and the academic editor and it was also seen again by the statistical reviewer. I am pleased to say that provided the remaining editorial and production issues are dealt with we are planning to accept the paper for publication in the journal.

[LINK]

We look forward to receiving the revised manuscript by Feb 24 2022 11:59PM.   

Sincerely,

Callam Davidson, 

Associate Editor 

PLOS Medicine

plosmedicine.org

Requests from Editors:

Lines 52, 79, and 443: Please update to ‘many food and beverage brands worldwide’. 

Line 71: Please add ‘(HR)’.

Figures 2 and 3: The legends for these figures do not explain the asterisks in the figures.

Lines 339-349: Please consider whether some of this content would be better placed in the Discussion (presentation of results from the previous study on lines 339-341 and the interpretation at lines 347-348). Presentation of the data in the Supplementary Materials should however remain in the Results section. 

Supplementary Results: I could not find reference to this part of the Supplementary Materials in the manuscript. 

Comments from Reviewers:

Reviewer #2: Alex McConnachie, Statistical Review

I thank the authors once again for considering my comments, and I am happy with the modifications they have made. I have no further comments.

[LINK]

---

## [Editor Report · Decision Letter 4]

23 Feb 2022

Dear Dr Debras, 

On behalf of my colleagues and the Academic Editor, Dr Wei Zheng, I am pleased to inform you that we have agreed to publish your manuscript "Artificial sweeteners and cancer risk: results from the NutriNet-Santé population-based cohort study" (PMEDICINE-D-21-04643R4) in PLOS Medicine.

PRESS

Sincerely, 

Callam Davidson 

Associate Editor 

PLOS Medicine